# Community engagement in health services research on soil-transmitted helminthiasis in Asia Pacific region: Systematic review

Cho Naing [1☺]*, Wong Siew Tung [2], Norah Htet Htet [2], Htar Htar Aung [2], Maxine A. Whittaker [1☺]*

**1** Division of Tropical Health and Medicine, James Cook University, Queensland, Australia, **2** School of Medicine, International Medical University, Kuala Lumpur, Malaysia

☺ These authors contributed equally to this work.
* cho3699@gmail.com (CN); maxine.whittaker@jcu.edu.au (MAW)

**Data Availability Statement:** All data generated or analysed during this study are included in this published article and its supplementary information files.

## Abstract

The research question was what studies are available that have assessed community engagement in the health services research on soil-transmitted helminths? We aimed to synthesise evidence on how communities were engaged in health services research on soil-transmitted helminths in low-and-middle-income countries of the Asia-Pacific Region. We focused on this region because soil-transmitted helminths are endemic, and their burden is significant in this region. This review was conducted according to the Preferred Reporting Items for Systematic Review and Meta-analysis (PRISMA) checklist. Relevant studies were searched in health-related databases including PubMed, Ovid, and Google Scholar. We selected studies based on the selection criteria set for this review. We collected textual information about the type of health services research, the degree of community engagement, the research phases involved, and the barriers/enablers affecting community engagement in research since they are pertinent to our review question and objective. Ten studies from seven countries in the Asia Pacific region were identified for this review. Albeit with variation in the extent of their involvement, various forms of communities/groups within communities were included such as Aboriginal communities, local communities, school children and their parents, school teachers and headmasters of schools, heads of villages, religious leaders, and so on. Overall, community engagement in health services research focused on of soil-transmitted helminths was limited. Six studies (60%) had collaboration at 'developing methodology', mainly through an explanation of the objectives of the study or study process to be conducted. Seven studies (70%) revealed community participation in soil-transmitted helminths at the "data collection" stage. Only one study (10%) documented that a community leader was involved as a co-author, reflecting an involvement in 'report writing' and further 'dissemination'. Findings suggest that there were various forms of community engagement in various aspects of the health services research context. Overall, there was moderate level of participation, but there was insufficient information on the partnership between various stakeholders, which prevented in-depth analysis of the engagement. Future health services research on soil-transmitted helminth interventions needs to be carefully planned, well

**Funding:** This work was supported by TDR (the Special Programme for Research and Training in Tropical Diseases, co sponsored by UNICEF, UNDP, the World Bank and WHO) [Project ID AP21-00287]. CN, MAW, NHH, HHA, WST are awarded this grant. None of these authors received salary from any commercial industries. The funding body has no role in the design of the study and collection, analysis, and interpretation of data and in writing the manuscript.

**Competing interests:** The authors have declared that no competing interests exist.

designed, grounded in principles of community engagement, and designed methodologically to allow in-depth participation by communities in all stages of the research.

## Introduction

The occurrence of neglected tropical diseases (NTD is closely related to risk factors including unsafe water, inadequate hygiene and sanitation, and poor housing conditions [1], and the health and economic burdens falls mostly on the poorest people and communities [2]. In 2020, the WHO estimated that more than 1 billion people were affected by NTDs, the majority of whom live in low- and middle-income countries (LMICs) [3]. The limited attention paid to NTDs globally and within country health programmes is partly due to their low mortality and that they often present with only subtle symptoms that hamper assessment of their burden [4]. Of these, soil-transmitted helminth infection (STH) represents the most prevalent cluster of NTDs worldwide [5] and receives less attention than would be suggested by the burden it takes on people.

The Alma Ata Declaration in 1978 framed community participation as critical in successfully delivering primary healthcare [6]. The global road map for NTDs 2021–2030 [3] highlight community engagement (CE) as a key enabler and intervention to address NTDs. Several national and international bodies recognize the importance of CE in research including the Bill and Melinda Gates Foundation [7], the Human Health and Heredity in Africa Initiative (H3Africa) [8], the National Institute of Health (NIH) and the Center for Diseases Control and Prevention (CDC) [9].

Public health programmes and policies are formulated at national and regional levels, while prevention and intervention usually take place at the community level. Conventionally, health interventions and actions were driven by professionals with little or no input from the targeted communities [10]. However, there are growing concerns over the limited application of research findings to inform health practices and policies targeted to the end users. Published studies documented that CE in health services can 'give a voice to the voiceless' [11], improve the acceptability of the intervention are more likely to result in contribute positive health outcomes [12], results in more efficient use of resources, strengthen coordination at local levels and build local capacities [13].

CE in research has been defined as "*a process of inclusive participation that support mutual respect of values, strategies, and actions for authentic partnership of people affiliated or self-identified by geographic proximity, special interest, or similar situations to address issues affecting the well-being of the community of focus*" [14]. Terms that are often used to describe CE in research include community-based participatory research, participatory action research [15], or collaborative partnership [7].

Health systems and services remain largely designed, implemented, and evaluated through an expert-driven, top-down process without recognising the role of CE as key factor of improving health and well beings [16]. This includes a significant proportion of the health services research (HSR) defined as "*a multidisciplinary field of inquiry, both basic and applied, that examines access to, and the use, costs, quality, delivery, organization, financing, and outcomes of health care services to produce new knowledge about the structure, process, and effects of health services for individuals and populations*" [17].

On the whole, the key question addressed in this review was: *what studies are available that have assessed the CE in HSR on STH*? Hence, we aimed to synthesise evidence on how

communities were engaged on research on STH in LMICs of the Asia-Pacific Region. We focused on this region because STH are endemic, and their burden was significant in this region. Understanding when, how and for whom CE can be effective in developing health service and system responses to STH is vital to inform efforts to meet the Sustainable Development Goals (SDGs).

## Methods

The reporting of this systematic review adhered to the Preferred Reporting Items for Systematic Review and Meta-analysis (PRISMA) (S1 Checklist). The current study was a part of larger study supported by TDR (the Special Programme for Research and Training in Tropical Diseases, co-sponsored by UNICEF, UNDP, the World Bank and WHO) [Project ID AP21-00287]. A protocol including this part of study was approved by the Ethics Review Committee of the International Medical University in Malaysia (ID: IMU R 272/2021). Formal consent was waved as the study included only published data, but not any human or animal participants. Hence, formal consent was not necessary to obtain.

### Study search

Relevant studies were searched in the health-related databases including PubMed, Ovid, Google Scholar, Cochrane Collaboration Library, Database of Abstract of Reviews of Effectiveness. The keywords with appropriate Bolen operators were used: "community engagement" "community participatory" "action research" "participatory research" "participatory action research" "community-based research" "soil transmitted helminths" " hook worms" " ascariasis" "trichuriasis". The search was extended to System for Information on Grey Literature including Social Science Research Network and Evidence for Policy Practice Information and Coordinating Centre (EPPI-Centre), Regional Bibliographic databases (e.g. Australian Education Index, AEI https://www.acer.org/my/library/australian-education-index-aei). Search terms for Ovid are available in S1 Table. The search was limited to English language publications between 1995 and June 2022 in LMICs in the WHO-Asia-Pacific region.

### Eligible criteria

Individual studies were selected based on the PECOS format.

**Participants (P)**: Communities residing in LMICs in WHO defined Western Pacific and Southeast Asia, regardless of age and gender. Communities is as defined in the primary studies.

**Exposure (E)**: Program/intervention targeting the health services need within the areas of STH, which involves community/stakeholder involvement or engagement and provide mechanisms and/or processes of community engagement.

**Context (C)**: Community- based and/or primary health care settings.

**Outcome (O)**: At least one health outcomes (i.e., self-efficacy/self-esteem/ self-regard, behavioral change, beneficial effects), and/or process evaluation (i.e., aiming to understand the functioning of an intervention by examining implementation, mechanisms and contextual factors.

**Study Design (S)**: Any study designs that described community-based interventions or study thereof on STH.

Studies were excluded, if they did not include participants from LMIC in the Asia Pacific region. Hence, descriptive studies, knowledge/attitude/practices studies, experimental clinical interventions with clinical outcomes or epidemiological studies that focus on the distribution of diseases were excluded.

## Data collection

Two investigators (NHH, CN) independently selected the included studies using pretested data collection sheets.

The two investigators independently collected the following textual data relevant to our review questions and objective:

- study setting, e.g., the geographical setting, the social, and cultural context.

- interventions e.g., type of intervention, how it was delivered, where it was implemented and by whom, funding sources, technical details, and mechanisms targeted by the intervention,

- description of the HSR: e.g., structure, process, outcome assessment

- extent of participants e.g., informed, collaboration, leading, developing the idea, developing methodology, data collection and writing, report writing, dissemination.

- facilitators/barriers encountered.

Any disagreements in these steps were settled by discussion with the third review author (MAW/WST).

## Data synthesis

Descriptive statistics were undertaken for the important variables, and reported as frequency (percentage) for categorical data, and mean (standard deviations, SD) for continuous data. We planned to do meta-analysis, if two or more studies provided numerical data of similar outcomes. meta-synthesis. However, a paucity of data in the selected studies precluded to do so.

**Assessment of the methodological quality.** The methodological quality of the included observational studies was evaluated using the "Risk of Bias in Non-randomised Studies-of-Interventions" (ROBINS -I) tool [18]. The ROBINS-I tool assessed seven domains of bias due to confounding, the selection of participants in a study, the measurement/classification of interventions/exposures, due to deviations from intended interventions/exposures, due to missing data, the measurement of the outcomes, and the selection of the reported results. For randomised clinical trials, the risk of bias in four domains such as random sequences generation, allocation concealment, blinding of participants and blinding of outcome assessors were evaluated [19]. Two investigators (WST, CN) independently assessed the methodological quality and any discrepancies were settled by discussion.

*Assessment of the level and extent of community engagement in HSR.* For this review, we did not evaluate the methodological quality (risk of bias) of the included studies because of the variations of their contents (i.e., design of studies, type of communities, features of the implemented interventions, and their goals). Instead, we assessed the level and extent of CE (i.e. leading, collaborating, consulted, informed, not informed/unclear), adapted from Brunton and colleagues [20].

In each level, the two investigators independently rated across research phases such as 'developing ideas', 'developing methodology', 'data collection/analysis, 'report writing' and 'dissemination. We gave 1 (+) score to 'leading' or 'collaborating', while 0 scores was given to no information/informed/consultation, as described elsewhere [21, 22]. Hence, the highest

score that a study can achieve was five. As an example, study X showed CE as 'consultation' at 'developing ideas', collaboration at 'developing methodology', and 'informed' at data collection/analysis, but 'no information' at reporting writing and 'collaboration' at dissemination stage, achieved a total score of 2 (i.e., 0+ 1+ 0 + 0 + 1 = 2). The quantum of engagement across all elements of the study was totaled to determine the extent of CE, and the overall extent was classified as high extent (score 4–5), moderate extent (scoring 2–3), and low extent (score 0–1). This score was developed by the researchers based on our experiences in previous study.

## Results

Fig 1 shows the study selection process. Initial searched yield 337 studies, including 121 duplicates, and 58 were not relevant to this review. After screening titles and abstracts, and removal of duplicates and irrelevant studies, 158 were checked. Further removal of 124 studies, and 34 full-text studies that had been initially assessed. Of them, ten studies across seven countries in the Asia Pacific region were eligible for this review [23–32]. The reasons for the exclusion of 24 studies are provided in S2 Table.

Table 1 provides the main characteristics of studies identified for this review. These included studies were published between 2008 and 2021. Three studies (3/10, 30%) were conducted in Philippines [25, 26, 30], while two studies (20%) in Malaysia [23, 24], and one study each in Cambodia [28], India [31], Indonesia [29], Laos [32], and the Solomon Islands [27].

Regarding the study design, three studies (30%) were cross-sectional design/survey [27, 28, 32], eight single studies were an RCT [23], a repeated screening [24], a post- evaluation of STH program [25], phase I/ II assessment [26], a qualitative exploratory study [30], a pilot intervention study [29] and a cluster RCT [31], respectively. All except one study [26] reported their funding sources. There was a diversity of funding sources that supported CE in STH research including TDR/WHO [27], Bill & Melinda Gates Foundation [31], pharmaceutical industries [25, 32], charitable foundation [29], and local institutional and/or national research grants [23, 24, 28]. Overall, the methodological quality of the studies identified for this review was of moderate level. The main methodological issues were "selection bias" and "classification bias" in observational studies, and a lack of adequate information on allocation concealment and blinding in RCTs (Fig 2).

### Community engagement hub

Amongst ten studies included in this review, there were various types of communities involved with variation in the extent of their involvement (Table 1). The participating communities included Aboriginal communities [23, 24], local communities [29] school children and their parents, and sub-groups within these communities such as, school teachers, headmasters of schools, heads of villages [24, 28, 32], local health staff [23, 29] monks [32], voluntary health workers (VHWs) [32], people working in non-health sectors, other government departments such as department of education (DepEd) [25], not for-profit organizations (i.e., Youth Union, Lao Women Union) [32] as well as in the private sector such as animation design experts from animation production companies [23].

CE was heterogeneous across the included studies, and communities' participations was only in some processes of the research cycles (Fig 3). It was noted that none of the included studies had communities leading/collaborating at 'developing ideas' stage. The communities in the studies were involved through being informed the objectives of the study before the commencement of the study. Six studies (60%) had collaboration at 'developing methodology' stage [23–26, 30, 31], again mainly through the provisions of an explanation about the objectives of the study or study process. Seven studies (70%) described community participation on

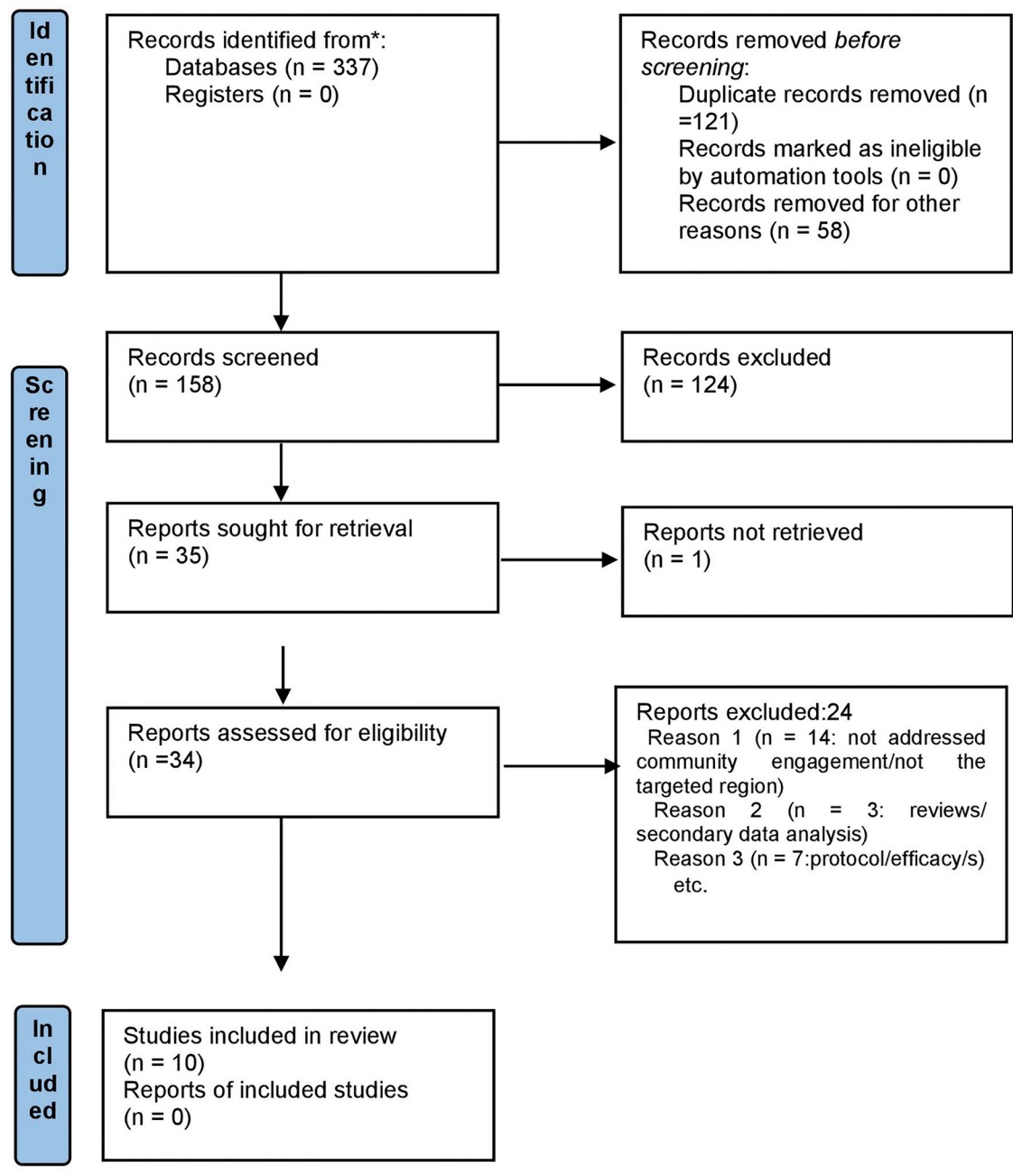

**Fig 1. Study selection process.**

STH at 'data collection' stage [23, 25–28, 30, 32]. One study (10%) documented that a community leader was involved as a co-author [27], reflecting an involvement in 'report writing' and further 'dissemination' stage.

In reflecting upon any engagement of communities in the research, some studies described a range of different mechanisms. For instance, the majority (60%) was CE through 'small community meetings' [23–25, 29, 30, 32] inclusive of stakeholders such as village heads [24, 25, 29, 30, 32], parents of school children [23], and school teachers [23, 24]. Other mechanisms were: provision of 'trainings' to community members [26] including VHWs [26], local government

**Table 1. Characteristic of included studies.**

| | Study [ref#] | Study period | Country | Intervention | Study design | Participants | Community engagement in stages of the research cycle | Mechanisms of engagement | Funding |
|---|---|---|---|---|---|---|---|---|---|
| 1 | Al-Delaimy, 2014 [23] | April-October 2013 | Malaysia, Pahang state | HELP | open-label controlled intervention trial for impact assessment | N = 317 SC from 2 primary schools in Orang Asli communities | collaboration with health, educational departments, LGU, commercial industry | **group discussions** with experts[1] from the field of Parasitology, PH, Education, Psychology, SH, Department of Orang Asli Development, education office, other researchers, SC, parents, teachers, health clinic personnel, animation design experts & animation production companies; **workshop** for teachers, teacher's guide book on STH, posters, a comic book, drawing activities, a sanitary bag, puppet show, nursery song videos. | University of Malaya High Impact Research Grant UM-MOHE UM.C/625/1/HIR/MOHE/MED/ 18 from the MOHE Malaysia & University of Malaya Research Grants; RG439/12HTM & PJM-KTP Community Project grant (FL001-13SBS). |
| 2 | Al-Mekhlaf, 2008 [24] | July 2006 – Feb 2007[a] | Malaysia, Pahang state | MDA for STH & schistosomiasis. | repeated screening | N = 120 SC from 18 Orang Asli villages; headmaster of the school, teachers, LCS, HO villages, parents of the SC | **Informed** explanation about the community involvement & objectives of study | Small community **meetings**\*; headmaster of the school, teachers, clinic staff, HO villages, parents, & their SC to give an explanation about their involvement & objectives of the study. | University of Malaya (ID P0101/2006B & PS178/2007B) |
| 3 | Belizario, 2014 [25] | NA | Philippine, Capiz Provinces | MDA for STH | pre-post evaluation of LGU-led SB teacher-assisted MDA | SC, LGU, DepEd, DOH, & Pharmaceuticals companies ** initiated MBL donation PG; Defining the objectives of the PG & responsibilities of stakeholders | LGUs; **collaboration** with health & educational agencies. | **cooperation** b/t the health & education sectors through FSP; **Local legislation** ensured continued local support for the control PG; **Advocacy/SM** initiated to ensure LGUs; **collaboration** with health & educational agencies. | Johnson & Johnson/ Janssen Pharmaceutic |
| 4 | Belizario Jr, 2015 [26] | | Philippines, Central & Southern | Integrated Helminth Control PG | Phase I assessed safety/feasibility of combined MDA; Phase II assessed feasibility of teacher-assisted combined MDA | N = 408 SC | training LCS, school teachers VHWSs, teachers & parents | **Training** for health personnel from the DepEd, LHU (MOs, nurses, midwives, VHWs); clinic teachers & LCS performed roll-out orientation of all class teachers/parents; Trained class teachers co-administered ALB/PZQ; Teachers accomplished record forms documenting coverage rate/ duration of combined MDA. The DOH provided the drugs and support for feeding prior to MDA, while the DepEd provided the infrastructure & manpower, LHUs provided local health workers, & the project team from the academe provided technical support; advocacy, capacity building, social mobilization, & monitoring/ evaluation assigned to concerned stakeholders i | NA |
| 5 | Bradbury, 2018 [27] | August 2014 | Solomon Islands, East Kwaio, | outreach MDA with ALB | survey | community research consortium of local health leaders, LCS from AAH & researchers from JCU & Central UQ | local health leaders, staff from AAH; **Community leader involved as a coauthor** | **community meetings** each night to update community members on the number of specimens examined & prevalence & intensity of STHs (p.193). | TDR/WHO (grant 1-811001688), JCU Development Grant (title: Elimination of STHs, One Village at a time) & Tropical Health Solutions. |

(*Continued*)

**Table 1.** (Continued)

| Study [ref#] | Study period | Country | Intervention | Study design | Participants | Community engagement in stages of the research cycle | Mechanisms of engagement | Funding |
|---|---|---|---|---|---|---|---|---|
| 6 Collela 2021 [28] | June–July 2016 | Cambodia | single dose of ALB | community-wide cross-sectional survey & screening using SFF & mqPCR | n = 1232 villagers | HO the villages, **local authorities** of province, CNM MOH Rovieng HC, | **HO the village was informed** about the study & provided information to the villagers, Administration of ALB by **directed observation of trained** field workers from MOH | Faculty of Veterinary & Agri- cultural Sciences Strategic Research Funds, The University of Melbourne. |
| 7 Gray 2019 [29] | July 2011 to June 2012 | Central Java, Indonesia | anti-helminthic drugs, construction, adoption of improved latrines, effective education regarding hygienic, & sanitary behaviour | Pilot study of integrated intervention | N = 527 villagers from two villages, Palemon & Cepoko, in the Gunungpati sub-district of the city of Semarang, Central Java | BALatrine is made **by local people** using local materials; householders themselves to take ownership over the latrines & their maintenance | Community HE programme was delivered via **community meetings** in each village through a two-hour village-wide mobilization meeting; a series of **small group workshops** took place with the villagers; | UBS Optimus Foundation |
| 8 Lorenzo 2019 [30] | 2013–2015 | Philippines, Northern Samar & Sorsogon | MDA program | qualitative exploratory study | DOH collaborated with DepED; teachers, SC & mothers; invited for data collection | led by teachers & principals with the supervision of health workers during first/third quarter of school yr | **Orientation** and HE to parents & SC; **Secured** medicines from DepED, health Offices; **Coordination** with target barangays, schools & pre-testing the data collection tools | DOH through RITM |
| 9 Patil,2014 [31] | May–July 2009 | India. Madhya Pradesh | TSC | cluster RCT, a part of multicountry study | state government, village community | **Informed** study objectives, use of collected information, confidentiality, risks, benefits, & respondent rights, etc | a series of **meetings** & site visits' a community award" NGP" to communities | Bill & Melinda Gates Foundation |
| 10 Phongluxa 2015 [32] | June 2007 onwards | Lao | CD intervention against liver fluke & STH | cross-sectional longitudinal study, a combination of quantitative & qualitative methods | HO village, HO Youth Union, HO Lao Women Union, monk, teacher, trained LCS monitored & supervised trained village leaders regarding the MDA; trained village group officer encouraged & promoted the anti-liver fluke intervention in community. | **involved in the data collection** HO village, HO Youth Union, HO Lao Women Union, monk, teacher for FGDs & IDIs; **trainings** of CDDs provided by Provincial & DHO; **monitoring & supervision of** village leaders; trained village leaders **returned the treatment records &** remaining drug to the DHO | **Trainings** to CDDs to perform their task well on the whole during the intervention | Give2Asia; Johnson & Johnson (Philippines) Inc. and its division, Janssen Pharmaceuticals (donating the MBL tablets forMDA |

[a], 7 months from the beginning of July;

* before commencement of the study;

** Johnson & Johnson Pharmaceuticals Philippines, Inc, and Janssen Pharmaceuticals Philippines, Inc;

AAH; Atoifi Adventist Hospital; ALB: albendazole; CB: community-based; CD: Community-directed; CDD: community drug distributor; Dec: December; CNM: Centre for Parasitology; Entomology and Malaria Control; DHO: District health office; DepEd: Department of Education; Feb: February; FSP: Food-for School Program of the Department of Education; HC: health centre; HE: health education; HELP: health education learning package; HO: head of: JCU: James Cook University; LCS: local clinic/health staff: LGU: local government unit; LHU: local health unit; MBL: Mebendazole; MDA: mass drug administration; MOH: ministry of health; mqPCR: multiplex qPCR; NA: not available/not reported; NGP: Nirmal Gram Puraskar; QoL: quality of life; RCT: randomized controlled trial; RITM: Research Institute for Tropical Medicine in Philippines; SB: school-based; SC: schoolchildren; SFF: standard faecal flotation; TSC: Total sanitation campaign; UQ: university of Queensland; VHW: village health workers;

| | Random sequence generation (selection bias) | Allocation concealment (selection bias) | Blinding of participants and personnel (performance bias) | Blinding of outcome assessment (detection bias) | Confounding | Selection bias | Classification bias | Bias due to deviations from intended interventions | Bias due to missing data | Outcome measurement bias | Reporting bias |
|---|---|---|---|---|---|---|---|---|---|---|---|
| Al-Delaimy, 2014 | ? | ? | − | − | | | | | | | |
| Al-Mekhlaf, 2008 | | | | | ? | + | + | ? | + | + | + |
| Belizario, 2014 | | | | | ? | + | + | ? | + | ? | + |
| Belizario Jr, 2015 | | | | | ? | + | + | ? | + | ? | + |
| Bradbury, 2018 | | | | | ? | + | + | ? | + | + | + |
| Collela, 2021 | | | | | ? | + | ? | + | + | ? | ? |
| Gray, 2019 | ? | ? | ? | ? | ? | + | + | + | + | ? | ? |
| Lorenzo 2019 | | | | | ? | + | + | ? | + | + | + |
| Patil, 2014 | + | ? | ? | ? | | | | | | | |
| Phongluxa 2014 | | | | | ? | + | + | ? | + | + | + |

Yellow: 'moderate risk of bias' in observational study or 'unsure' risk of bias in RCT;

Green: low risk of bias; Red: high risk of bias.

**Fig 2. Risk of bias assessment.**

staff [26], community drug distributors [32], and provision of 'infrastructure and human resources' (school teachers/headmasters) from DepEd [26, 30]. Overall, the majority of studies (8/10, 80%) were rated as having a moderate level of CE [23, 25–30, 32], and two studies demonstrated at low level of CE [24, 31].

| No. | Study [ref#] | Country | Developing ideas | Developing methodology | Data collection/analysis | Report writing | dissemination | Scoring | level |
|---|---|---|---|---|---|---|---|---|---|
| 1 | Al-Delaimy, 2014 [23] | Malaysia | informed | colloboration | colloboration | not involved | collaboration | 0+1+1+0+1=3 | Moderate |
| 2 | Al-Mekhlaf, 2008 [24] | Malaysia | Informed | colloboration | not involved | not involved | not involved | 0+1+0+0+0=1 | Low |
| 3 | Belizario, 2014 [25] | Philippine | informed | collaboration | collaboration | not involved | not involved | 0+1+1+0+0=2 | Moderate |
| 4 | Belizario, 2015 [26] | Philippine | involved | collaboration | collaboration | not involved | not involved | 0+1+1+0+0=2 | Moderate |
| 5 | Bradbury, 2018 [27] | Solomon Islands | informed | consultation | collaboration | collaboration | collaboration | 0+0+1+1+1=3 | Moderate |
| 6 | Colella, 2021 [28] | Cambodia | informed | consultation | collaboration | collaboration | collaboration | 0+0+1+1+1=3 | Moderate |
| 7 | Gray, 2019 [29] | Indonesia | informed | consultation | collaboration/leading | collaboration | collaboration | 0+0+1+1+1=3 | Moderate |
| 8 | Lorenzo, 2019 [30] | Philippine | informed | collaboration | collaboration | not involved | not involved | 0+1+1+0+0=2 | Moderate |
| 9 | Patil, 2014 [31] | India | informed | collaboration | not involved | not involved | not involved | 0+1+0+0+0=1 | Low |
| 10 | Phongluxa, 2014 [32] | Lao | informed | consultation | collaboration | collaboration | not involved | 0+0+1+1+0=2 | Moderate |

**Fig 3. Level and extent of community engagement in various phases of HSR (N = 10 studies).** The role of communities in HSR is ranged from red (lowest level) to green (highest level). Consultation, informed or not involved (Red): score 0; Leading or collaboration (Green): score 1 (adapted from Brunton 2015 [21–22]).

## Facilitators and barriers (Table 2)

Seven studies in this review (7/10, 70%) addressed the ranges of drivers that facilitated CE in HSR targeting STH [23, 25, 27–32]. For instance, communities were motivated to engage through local legislation that could further ensure budgetary support [25], incentives such as awards to communities [31], 'conditional cash grants' [30], providing treatment drugs free of charge [32], multisectoral collaboration among stakeholders [23, 26], use of inexpensive local materials [29] and social mobilization [31]. As an example, the requirements of the *Pantawid Pamilyang Pilipino* program (4 Ps), a government program was that it provided 'conditional cash grants' to extremely poor families to ensure that children receive deworming pills twice a year [30]. A study with Orang Asli communities (Aboriginal) in Malaysia stated that "involvement of teachers in such a programme is perceived as being crucial in order to achieve the sustainability and efficiency of the control programme" [23]. However, in the same study it noted that there was concern that involvement of teachers may disturb other forms of school-based activities, and may be an additional burden on the teachers [23].

Studies in the Philippines highlighted that building confidence among school teachers and health workers to implement combined mass drug administration (MDA) facilitated their involvement [25, 26]. These studies also described how multisectoral stakeholder cooperation helped to conserve funds for the initial assessment and management of adverse events (i.e., assurances and rest) by using local health workers provided with sufficient orientation and training rather than using medical doctors [25, 26].

Amongst the barriers stated, common was community members' doubts about the effectiveness of deworming drugs, and their potential side effects [30], leading to their less involvement. Another barrier to CE was related to the communities' perceptions of the low quality of

**Table 2. Barriers and facilitators.**

| Study, year | Targeted outcome | Facilitators | Barriers | Remarks/recommendation |
|---|---|---|---|---|
| Al-Delaimy 2014 [23] | Impact assessment of HELP | Involvement of teachers in a PG is perceived as being crucial in order to achieve the sustainability & efficiency of the control PG | people are unable to purchase more than one pair of shoes, & in some cases may not even be able to afford; most of these children had only school shoes & their parents did not allow them to use these shoes when moving around their villages.; toilets were used as storage rooms due to cultural beliefs that toilets should not be located inside the house; SB activities may disturb other forms of schooling & may add a burden to the teachers; | improving and integrating the package in to the overall curriculum & normal school activities.; poverty & low levels of education require more direct intervention by the government |
| Al-Mekhlaf, 2008 [24] | Repeated screening of STH | NA | the adherence of these people to be confined within the jungle constraining the strategy. | innovative long-term interventions such as providing job opportunities & improving quality of education; to reduce the poverty & improve the QoL of Aboriginal communities |
| Belizario, 2014 [25] | MDA for STH | LGUs, teachers in MDA, local legislation ensured continued local support for the control PG/ budgetary support; participation of teachers in MDA contributed to the success of PG;. | NA | |
| Belizario Jr, 2015 [26] | MDA for STH | referral system linking schools to the LHUs & referral hospitals; LHS & education personnel participated in the implementation of combined MDA in an existing infrastructure; Administration of a combination of two anthelminthics to SC by teachers, with supervision from local nurses or midwives allowed achievement of higher coverage; multisectoral collaboration among stakeholders. | NA | SB teacher-assisted combined MDA may be further scaled up to implement in other co-endemic provinces or areas |
| Colella 2021 [28] | survey & screening of STH | enrolling individuals that had to voluntarily present themselves to the pagoda might have attracted people that were more compliant with hygiene & deworming practices, | NA | refined guidelines are needed for a sustained reduction in morbidity based on local epidemiological knowledge of the dominant species |
| Gray 2019 [29] | anti-helminthic drugs, construction, adoption of latrines | squat latrine that can be constructed by village residents using inexpensive materials, by being culturally familiar, simple, cheap, easy to build & maintain & adaptable for wet & dry conditions. | old habits of open defecation or cultural preferences can be difficult to overcome. | sanitation interventions can be effective at reducing worm burden when designed appropriately for the local context & combined with HE & promotion. |
| Lorenzo, 2019 [30] | MDA for STH | "4Ps", a government PG that provided conditional cash grants to extremely poor families. | misconceptions of drug effectiveness: • mothers refuse signing the consent forms if their SC were sick; • doubts over the effectiveness of the pills provided in schools; • believed that store-bought drug was more effective, because the dead worms came out of children's body, compared to the effects of MDA drugs (provided by the national government), where worms would crawl out alive; " the government provided the toilet bowl but what about the money for cement? They should provide complete materials, like hollow blocks, & cement". | strengthening HE messages, & increasing visibility or availability of on-site medical personnel. |

*(Continued)*

**Table 2.** (Continued)

| Study, year | Targeted outcome | Facilitators | Barriers | Remarks/recommendation |
|---|---|---|---|---|
| Patil,2014 [31] | Total sanitation campaign | ongoing SM & behavior change activities at state/district/village levels; flexible technology options for toilet; a community award "NGP" to communities that were determined to be "open defecation free"; | imperfect compliance with treatment assignment; poor fidelity of intervention implementation; a gap b/t the planned follow-up period and the actual follow-up measurement (18 M vs 21 M) | More SM & exposure to behavior change activities & IHL construction in intervention villages compared to control village. |
| Phongluxa 2015 [32] | CD intervention against liver fluke & STH | encouraged community members to get treatment which is FOC & directly in the village | traditional way of raw fish dish consumption raw or insufficiently cooked fish dishes; community leaders have not yet acquired proper knowledge about MDA PG | HE in the communities is the most suitable approach |

BPL: below poverty line; b/t; between; FOC: free of charge; HE: health education; HELP: health education learning package; IHL: individual household latrines; LGUs: Local government units; M; months; MDA: mass drug administration; NGP: Nirmal Gram Puraskar; PG: program; QoL: quality of life; SE: socioeconomic status; SM: social mobilization; "4Ps": Pantawid Pamilyang Pilipino Program;SB:school-based; SC: school children; SM:social marketing; STH: soil-transmitted helminth;

the government supplied drugs in the MDA compared to commercially available anthelminthics. For instance, a female participant in mass deworming stated, "*I feel like the drug (albendazole) does not automatically kill the worms because some would still come out alive. Unlike with Combantrin (the store-bought drug), worms passed with stool were actually dead*" [30]. Communities were finding it difficult to change previous practices of open defecation, according to a study on the use of low-cost sanitary latrines in Indonesia [29].

## Discussions

### Summary of evidence

The present review of ten studies from seven LMICs in the Asia Pacific Region revealed minimal/limited CE in the research. We found that the depth of community involvement ranged from the basic (i.e. information being provided to communities) to a collaborative partnership.

The majority of the studies that composed this review were designed to report on health outcomes, with no description or evaluation of the level of community involvement in HSR. This revealed that rather than focusing on the level of CE in HSR, the primary objectives of these primary studies were to investigate health consequences.

CE has been defined as "*the process of developing relationships that enable stakeholders to work together to address health related issues and promote wellbeing to achieve positive health impact and outcomes*" [33]. In the present review, communities were not limited to being defined as geographically constructed such as in villages, but included social groups united by activities as well.

### Comparison with the published reviews

Similar to a published review [34], communities are regulated by power relations such as local governments, which participated in community-based interventions [25, 32]. According to a published review [15], community participation in PHC and water resource governance in South Africa was likewise tokenistic, regarding the Arnstein's ladder of involvement [35]. Since consultations and/or informing were the most typical forms of participation, the

Arnstein's ladder of participation [35] was similarly appropriate to the degree of CE in our review of STH.

There was also limited description of the community engagement mechanisms utilised in the studies (e.g. for small scale community meetings with health staff, village heads, school headmasters, teachers, parents of school children, NGOs), and creating knowledge gaps in the field, and limiting the expansion of lessons learned in the field. The main mechanisms of CE in this evaluation were small meetings with community leaders, community members, and health personnel, similar to prior reviews that evaluated CE [15, 34–36]. This approach may help build trust between the researchers and the community, and facilitate partnership, as well as provide researchers with local contextual information about the community [8]. However, details about how these "meetings" were organised, managed, and how results from the meetings were incorporated into the research were not discussed in these included studies. It has been hypothesised that increased CE in HSR studies may allow for greater sustainability of the intervention programme e.g., through sharing of resources among health sector and non-health sectors (DepEd in this case), community stakeholders (Aboriginal peoples, NGOs in this case) or promoting collective ownership [37]. In fact, CE in research has gained popularity as an approach for enhancing research, ensuring that community concerns are taken into consideration, and it also inform ethical decision-making when research is undertaken in the context of vulnerability [38].

## Study limitations

Due to limited description of the extent of CE, lack of reporting the mechanisms for engagement and sustainability, it was difficult to assess the true extent of CE using the typology developed for this review. The studies included had objectives linked to health outcomes, and ignored non-health outcomes such as social benefits, reduction of inequities (SDG10) and strengthening partnerships (SDG17) [39].

## Conclusions

Findings suggest that various forms of community had engaged in various aspects of the HSR context. Overall, there was moderate level of participation in health services research aimed at controlling soil-transmitted helminths, but the remaining concerns were insufficient information on how the partnership developed between various stakeholders. Future health services research on soil-transmitted helminths interventions should be carefully planned, well designed, grounded in community engagement, with specific methods, context-appropriate, and in-depth participations.

## Supporting information

**S1 Checklist. PRISMA checklist.**
(DOC)

**S1 Table. Search strategy and information sources.**
(DOC)

**S2 Table. Excluded studies and reasons for exclusion.**
(DOC)

## Acknowledgments

We are grateful to the participants and researchers involved in the primary studies identified for this review.

## Author Contributions

**Conceptualization:** Cho Naing, Maxine A. Whittaker.

**Data curation:** Cho Naing, Wong Siew Tung, Norah Htet Htet, Htar Htar Aung, Maxine A. Whittaker.

**Formal analysis:** Cho Naing, Wong Siew Tung, Maxine A. Whittaker.

**Funding acquisition:** Cho Naing, Wong Siew Tung, Norah Htet Htet, Htar Htar Aung, Maxine A. Whittaker.

**Investigation:** Cho Naing, Wong Siew Tung, Norah Htet Htet, Maxine A. Whittaker.

**Methodology:** Cho Naing, Wong Siew Tung, Htar Htar Aung, Maxine A. Whittaker.

**Supervision:** Cho Naing.

**Writing – original draft:** Cho Naing.

**Writing – review & editing:** Maxine A. Whittaker.

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
