## [Decision Letter · Decision Letter 0]

7 Dec 2022

PGPH-D-22-01639

Community Engagement in Health Services Research on Soil-Transmitted Helminthiasis in Asia Pacific Region: Systematic review

Dear Dr. Naing,

Thank you for submitting your manuscript to PLOS Global Public Health. After careful consideration, we feel that it has merit but does not fully meet PLOS Global Public Health’s publication criteria as it currently stands. Therefore, we invite you to submit a revised version of the manuscript that addresses the points raised during the review process.

Please review the peer review reports below closely, paying particular attention to the comments related to the methodology, and either make the suggested changes or provide further justification for the existing methodological choices as needed. 

We look forward to receiving your revised manuscript.

Kind regards,

Claire J Standley

Academic Editor

Journal Requirements:

Additional Editor Comments (if provided):

Thank you for submitting your article. Both peer reviewers have questioned aspects of the search methodology, including the choice/variety of key words as well as the exclusion criteria, in part as the current approach seems to have excluded several articles which appear to address the topic of community engagement in STH programs in the Asia-Pacific Region. Please ensure that these comments are robustly addressed, either through changes to the methodology or additional explanation and justification in the manuscript, when submitting a revised manuscript. Please also note the additional minor comments made by each reviewer.

Reviewers' comments:

Reviewer's Responses to Questions

**Comments to the Author**

1. Does this manuscript meet PLOS Global Public Health’s publication criteria? Is the manuscript technically sound, and do the data support the conclusions? The manuscript must describe methodologically and ethically rigorous research with conclusions that are appropriately drawn based on the data presented.

Reviewer #1: Yes

Reviewer #2: Yes

2. Has the statistical analysis been performed appropriately and rigorously?

Reviewer #1: N/A

Reviewer #2: No

3. Have the authors made all data underlying the findings in their manuscript fully available (please refer to the Data Availability Statement at the start of the manuscript PDF file)?

Reviewer #1: Yes

Reviewer #2: Yes

4. Is the manuscript presented in an intelligible fashion and written in standard English?

Reviewer #1: Yes

Reviewer #2: Yes

5. Review Comments to the Author

Reviewer #1: General: The review focuses on an important and increasingly prominent topic, namely the involvement of study communities in scientific studies, here those conducted in South-East Asia and dealing with soil-transmitted helminths. Only 8 relevant publications were identified. The manuscript is easy to follow and generally well written. This reviewer has not fundamental concerns about the study but offers a number of comments to further improve the manuscript:

- Line 78: “people and the act that is the most common cluster” – the second part of the sentence is unclear. Consider revising.

- Line 131: why was “trichuriasis” not included as keyword when “ hook worms” and “ascariasis” were?

- Line 141: “Asian” should be “Asia”

- Line 203: “0+ 1+ 0+0++1= 2)” should be “0+ 1+ 0+0+1= 2)”

- Table 1 and Table 3: check format – the chosen format is unsuitable for publication due to very long and narrow columns.

- Lines 309-315: it is unclear why this paragraph appears here as it does not contribute to the topic of interest: it focuses on barriers to MDA acceptance, not community involvement

- Line 328: do you mean 8 studies?

- Lines 332-335: the sentence is very long and unclear. Consider revising.

- Lines 340-341: the last sentence of the paragraph is unclear. Consider revising.

- Discussion: the actual discussion seems to be missing – there is only a “summary of evidence”, “study limitations” and “conclusions”. A point to discuss is e.g. a comparison of the findings with the situation in other areas of public health, and/or other geographical areas. Other possible points to discuss include the optimal design of studies in this area, considerations under which circumstances community engagement is especially critical/not feasible, and others.

Reviewer #2: This study reports important information on CE during STH research that can aid in better planning and research on NTDS in the Asia Pacific.

Major remarks:

Authors should refine/revise their search criteria; for example “hook worms” should be “hookworms”. Further, some studies with clear CE seem to have not been included in their analyses. For example, in https://www.thelancet.com/journals/lanwpc/article/PIIS2666-6065(21)00167-X/fulltext and in

https://www.mdpi.com/2414-6366/4/4/141

there is a clear CE, with head of the villages involved in the study. I would suggest author to re-perform their search. For example, the above study came up with a simple search of hookworm AND Asia AND community.

It is not clear why some exclusion criteria were chosen. For example, how some studies included in their analyses are different from those in table S2 is difficult to understand. Given the title of this manuscript, all research studies with a CE component in STH research should be included in the analyses. Some studies in S2 that were deemed as “efficacy” or “cross sectional” are nowhere different from those included in table 1.

Minor remarks:

The text needs a thorough revision (e.g., often brackets are opened but not closed).

Line 290: if out of 11, it shouldn’t be that %

Line 360: statement not needed

Line 368: missing a verb

6. PLOS authors have the option to publish the peer review history of their article (what does this mean?). If published, this will include your full peer review and any attached files.

**Do you want your identity to be public for this peer review?** For information about this choice, including consent withdrawal, please see our Privacy Policy.

Reviewer #1: **Yes: **Peter Steinmann

Reviewer #2: No

---

## [Decision Letter · Decision Letter 1]

16 Feb 2023

Community Engagement in Health Services Research on Soil-Transmitted Helminthiasis in Asia Pacific Region: Systematic review

PGPH-D-22-01639R1

Dear Professor Naing,

We are pleased to inform you that your manuscript 'Community Engagement in Health Services Research on Soil-Transmitted Helminthiasis in Asia Pacific Region: Systematic review' has been provisionally accepted for publication in PLOS Global Public Health.

Best regards,

Claire J Standley

Academic Editor

Please see Reviewer 2's comment about ensuring full access to the data and selection process for the articles analyzed in this systematic review - my understanding is that these details are provided in Supplemental Information table S2, but please verify during the final formatting/copyediting stages.

Reviewer Comments (if any, and for reference):

Reviewer's Responses to Questions

**Comments to the Author**

1. If the authors have adequately addressed your comments raised in a previous round of review and you feel that this manuscript is now acceptable for publication, you may indicate that here to bypass the “Comments to the Author” section, enter your conflict of interest statement in the “Confidential to Editor” section, and submit your "Accept" recommendation.

Reviewer #2: All comments have been addressed

2. Does this manuscript meet PLOS Global Public Health’s publication criteria? Is the manuscript technically sound, and do the data support the conclusions? The manuscript must describe methodologically and ethically rigorous research with conclusions that are appropriately drawn based on the data presented.

Reviewer #2: Yes

3. Has the statistical analysis been performed appropriately and rigorously?

Reviewer #2: N/A

4. Have the authors made all data underlying the findings in their manuscript fully available (please refer to the Data Availability Statement at the start of the manuscript PDF file)?

Reviewer #2: No

5. Is the manuscript presented in an intelligible fashion and written in standard English?

Reviewer #2: Yes

6. Review Comments to the Author

Reviewer #2: (No Response)

7. PLOS authors have the option to publish the peer review history of their article (what does this mean?). If published, this will include your full peer review and any attached files.

**Do you want your identity to be public for this peer review?** For information about this choice, including consent withdrawal, please see our Privacy Policy.

Reviewer #2: No
